# Perturbating Intramolecular Hydrogen Bonds through Substituent Effects or Non-Covalent Interactions

**DOI:** 10.3390/molecules26123556

**Published:** 2021-06-10

**Authors:** Al Mokhtar Lamsabhi, Otilia Mó, Manuel Yáñez

**Affiliations:** Departamento de Química (Módulo 13, Facultad de Ciencias) and Institute of Advanced Chemical Sciences (IadChem), Universidad Autónoma de Madrid, Campus de Excelencia UAM-CSIC, Cantoblanco, 28049 Madrid, Spain; otilia.mo@uam.es

**Keywords:** intramolecular hydrogen bonds, amino-alcohols, α-substitution, beryllium bonds, calculated infrared spectra

## Abstract

An analysis of the effects induced by F, Cl, and Br-substituents at the α-position of both, the hydroxyl or the amino group for a series of amino-alcohols, HOCH_2_(CH_2_)_n_CH_2_NH_2_ (*n* = 0–5) on the strength and characteristics of their OH···N or NH···O intramolecular hydrogen bonds (IMHBs) was carried out through the use of high-level G4 ab initio calculations. For the parent unsubstituted amino-alcohols, it is found that the strength of the OH···N IMHB goes through a maximum for *n* = 2, as revealed by the use of appropriate isodesmic reactions, natural bond orbital (NBO) analysis and atoms in molecules (AIM), and non-covalent interaction (NCI) procedures. The corresponding infrared (IR) spectra also reflect the same trends. When the α-position to the hydroxyl group is substituted by halogen atoms, the OH···N IMHB significantly reinforces following the trend H < F < Cl < Br. Conversely, when the substitution takes place at the α-position with respect to the amino group, the result is a weakening of the OH···N IMHB. A totally different scenario is found when the amino-alcohols HOCH_2_(CH_2_)_n_CH_2_NH_2_ (*n* = 0–3) interact with BeF_2_. Although the presence of the beryllium derivative dramatically increases the strength of the IMHBs, the possibility for the beryllium atom to interact simultaneously with the O and the N atoms of the amino-alcohol leads to the global minimum of the potential energy surface, with the result that the IMHBs are replaced by two beryllium bonds.

## 1. Introduction

The name “hydrogen bonding” was used for the first time by Linus Pauling in 1929 [1], although the interactions associated with a hydrogen bond had been described for the first time more than a century ago, by W.M. Latimer and W.H. Rodebush, in a paper published in 1920 [2]. Since 1934, the year in which Pauling and Brockway experimentally confirmed that carboxylic acids could indeed form hydrogen bonds [3], hydrogen bonding became one of the most fruitful concepts in chemistry, being behind a huge amount of stabilizing intermolecular interactions [4,5,6,7,8,9,10,11,12,13,14,15]. In addition, in the 1920s, the existence of intramolecular interactions that, nowadays, we know as intramolecular hydrogen bonds (IHBs), were described in the literature. Indeed, the first paper on the coordination of hydrogen to explain the abnormal solubilities of some benzene derivatives was published in 1924 [16]. The intramolecular hydrogen bond was actually described in a 1926 publication by H.E. Amstrong [17], making reference to a paper of T.M. Lowry on the anomalies of the optical rotatory dispersion of tartaric acid published the same year [18], as a “bigamous hydrogen”. Almost a century has elapsed since the publication of these articles, and the presence of IHBs is very often shown to be behind many different phenomena in chemistry, physics, in particular in photophysics, material science, and biochemistry, to the point that IUPAC dedicated a specific paper to the definition of this chemical interaction [19]. Let us mention here a few suitable examples, drawn from a huge number of publications, such as the substituent effects on the IHB of malonaldehyde [20], or the role of IMHBs on: protonation and deprotonation processes [21], NMR chemical shifts [22], bond dissociation enthalpies [23,24], or their influence on the role of aromaticity in chemical reactions [25]. They are also crucial to understand excited-state intramolecular proton transfers [26,27,28], or the control of photosubstitution phenomena in metal complexes [29], or the control of emissive properties of different organic compounds [30]. IMHBs are also behind the high chemical stability of metal-organic frameworks [31] and the excellent electroluminescence of some organic light-emitting diodes (OLEDs) [32], and related with linear and nonlinear electric properties [33] or with the thermodynamic properties of compounds like pyridinol derivatives [34]. In biochemistry, they are responsible for proton shuttle mechanisms in certain lipase-catalyzed reactions [35]. IMHBs between hydroxyl and amino groups of a serinol function, characteristic of an important lipid, sphingosine, seem to be responsible for its existence, as neutral at the physiological pH [36], IMHBs allow also for designing novel peptide inhibitors [37], and play an important role as far as the molecular conformation of amino acids is concerned [38]. In addition, important properties are associated with the so-called intramolecular charge-inverted hydrogen bonds [39,40].

Since the pioneering studies by microwave spectroscopy on 2-aminoethanol [41] and by infrared spectroscopy on *trans*-8a and -8b-decahydroquinilinol reveal the important role of IMHBs in their stability, the interest in amino-alcohols increased significantly. The first ab initio calculations showed that 2-aminoethanol, 3-aminopropanol, and 4-aminobutanol were indeed stabilized by OH···N IMHBs, this stabilization being larger the longer the chain [42]. Very recent microwave experiments in parallel with MP2 ab initio calculations confirmed that, in both 3-aminopropanol [43] and 4-aminobutanol [44], the ground state is stabilized through the formation of an OH···N IMHB with O-N internuclear distance of 2.856 and 2.954 Å, respectively. In addition, a recent and rather complete study [45] using FTIR measurements, ^1^H NMR spectroscopy, density functional theory (DFT) calculations, and molecular dynamics (MD) on 3-aminopropan-1-ol, 3-methylaminopropan-1-ol, and 3-dimethylaminopropan-1-ol, unambiguously showed that the methylation at the N atom results in a systematic enhancement of the OH···N IMHB, reflecting a decrease of the s-character of the nitrogen lone pair orbital.

These results clearly indicate that the effects of substituents directly attached to the active sites of the OH···N IMHB have a significant effect in its strength, but, to the best of our knowledge, not much is known on the effects induced by substituents that are at the α-position of both the hydroxyl or the amino group, or what the effects are when any of these two groups are also actively participating in a non-covalent interaction with a second body. Hence, one of the aims of this paper is to investigate, through the use of high-level ab initio calculations, the characteristics of the OH···N IMHB in the series of HOCHX(CH_2_)_n_CH_2_NH_2_ and HOCH_2_(CH_2_)_n_CHXNH_2_ (*n* = 0–5) with substituted amino- alcohols being X = H, F, Cl, Br. As mentioned above, there is an alternative way in which the strength of an IMHB can be altered, and this is through non-covalent interactions in which either the proton donor or the proton acceptor is engaged. Among the many possible non-covalent interactions capable of interacting with IMHBs, the so-called beryllium bonds [46] are particularly interesting. A good and rather complete compilation on the coordination chemistry of beryllium was reported recently by Perera et al. [47]. Beryllium bonds have been found to compete with dihydrogen bonds [48], and they are able to modulate the strength of tetrel bonds [49]. They can also be formed when interacting with other electron deficient systems, such as boron derivatives [50] or with noble gases [51]. Although there are previous studies on the mutual interaction of intermolecular hydrogen bonds and beryllium bonds [52,53], very little has been done dealing with IMHBs. In this respect, a recent theoretical study must be mentioned describing how intramolecular hydrogen bonds are able to enhance tetrel bonds [54], or the competition between pnictogen bonds and intramolecular hydrogen bonds in protonated systems [55]. Here, we have decided to investigate this phenomenon when the system interacting with the amino-alcohol is an electron deficient compound. To achieve this goal, we will investigate the characteristics of the complexes between HOCH_2_(CH_2_)_n_CH_2_NH_2_ (*n* = 0–3) when they interact, either through the amino group or the hydroxyl group, with BeF_2_ molecules, yielding the corresponding beryllium bonds [46].

## 2. Computational Details

In order to ensure the reliability of the theoretical characterization of the IMHB stabilizing the compounds under investigation, we have decided to use the Gaussian-4 (G4) theory [56]. The G4 theory is a high-level ab initio composite method based on DFT optimized geometries and thermochemical corrections obtained at the B3LYP/6-31G(2df,p) level [56]. Correlation effects are accounted for by using the Moller–Plesset perturbation theory up to the fourth-order and CCSD(T) coupled cluster theory. A further correction, to account for the Hartree–Fock basis set limit, is added using a linear two-point extrapolation scheme and quadruple-zeta and quintuple-zeta basis sets.

However, it would be impossible to apply the G4 scheme to characterize all possible conformers associated with these two series of substituted amino-alcohols HOCHX(CH_2_)_n_CH_2_NH_2_ and HOCH_2_(CH_2_)_n_CHXNH_2_ (*n* = 0–5, X = H, F, Cl, Br) under scrutiny because this number is huge in particular when n ≥ 3, and it would be applied only to the most stable ones. In order to make the selection of the most stable conformers, we first located the ensemble of them by using the conformer–rotamer ensemble sampling tool (CREST) recently developed by Grimme [57,58]. This method is based on the semiempirical tight-binding based quantum chemistry method GFN2-xTB [59] in the framework of meta-dynamics (MTD). Once the conformers ensemble is obtained, the script ENSO (Energetic Sorting of Crest ensembles) [60] is applied to classify them in a three-step process. The first one consists of a single point calculation at pbeh-3c/Def2-SVP [61] level of theory of the different structures of the ensemble. In a second step, the conformers in the range of 25 kJ/mol are selected for optimization at the same level of theory. In the last step, the range of energy is reduced to about 8 kJ/mol, the final energy, and the percentage of Boltzmann distribution being obtained from a single point wb97x/Def2-TZVPP calculations [62]. The conformers with a percentage higher than 1% were chosen to carry out the G4 calculations. It should also be mentioned that, for all the complexes including Beryllium bonds, final energies were also calculated at the G4 level, but, in these cases, instead of using the standard geometry optimization procedure within the G4 formalism, the structures were optimized using a larger basis set expansion, aug-cc-pVTZ, that includes diffuse functions, very often critical to correctly describe non-covalent interactions.

The bonding in all the systems investigated was analyzed through the use of four different approaches, namely the quantum theory of atoms in molecules (QTAIM) [63,64], the electron localization function (ELF) [65], the natural bond orbital (NBO) analysis [66], and the non-covalent interaction (NCI) formalism [67]. In the QTAIM approach, a topological analysis of the electron density of the systems permits the location of its critical points and the paths of minimum gradients connecting them, which leads to an unambiguous definition of chemical bonds, and/or ring and cage structures. The ELF procedure allows a partition of the molecular space in monosynaptic and disynaptic (or polysynaptic) basins in which the electrons of the system are distributed. The monosynaptic ones are associated with core electrons and/or lone pairs, whereas the disynaptic (or polysynaptic) correspond to bonding regions. The NBO method is based on the generation of localized hybrid orbitals which would correspond to Lewis-type representations of the molecular structures. The calculation of second-order orbital interaction energies between occupied and empty orbitals permit quantitatively characterizing donations and backdonations among occupied and empty localized molecular orbitals involved in inter and/or intramolecular interactions. The NCI formalism is an alternative analysis of the electron density, based on the fact that regions of small reduced-density gradient at low electronic densities are associated with the presence of non-covalent interactions. This analysis leads to rather visual representations if a colored scale is used to plot the isosurfaces of the reduced density, in both two- and three-dimensional spaces.

## 3. Results and Discussion

In our theoretical survey, we have investigated, for each one of the species under consideration, the possibility of having O–H···N or N–H···O IMHBs, though, in almost all cases, the former are stronger than the latter, and, therefore, in most of the cases, the global minima are characterized by the existence of an O–H···N IMHB.

The number of stable conformers of HOCHX(CH_2_)_n_CH_2_NH_2_ and HOCH_2_(CH_2_)_n_CHXNH_2_ (*n* = 0–5, X = H, F, Cl, Br) compounds obviously increases as the number of carbon atoms increases along the series, reaching for the larger values of *n* several hundred. In what follows, we will analyze the six more stable unsubstituted HOCH_2_(CH_2_)_n_CH_2_NH_2_ (*n* = 0–5) amino-alcohols first, and, after that, we will pay attention to the effect of the X substituent at the α-position with respect to the CH_2_OH group and with respect to the CH_2_NH_2_ group.

### 3.1. HOCH_2_(CH_2_)_n_CH_2_NH_2_ (n = 0–5) Compounds

We will start our analysis by looking at the effect that the length of the carbon chain has on the strength of the IMHB between the alcohol and the amino functional groups. As expected, the conformer stabilized by the formation of a O–H···N IMHB is more stable (see Appendix A) than that exhibiting a N–H···O IMHB because the hydroxy group is a better proton donor but a weaker base than the amino group, so, in what follows, we will discuss only the systems exhibiting a O–H···N IMHB.

The structures corresponding to the most stable conformer for each value of *n* are shown in Figure 1. It can be seen that the longest O–H···N IMHB corresponds to the first member of family, 2-amino ethanol (*n* = 0) which should be the amino-alcohol with the weakest bond along the series. This bond length reaches its minimum for *n* = 2, which in principle should be the member of the series with the strongest O–H···N IMHB, its strength decreasing for larger values of *n* (*n* = 0–5).

Unfortunately, although the calculation of the energy of an intermolecular hydrogen bond is straightforward, this is not so when dealing with IMHBs, though some procedures to estimate it have been proposed [68]. We propose, however, the use of the isodesmic reaction [69] (1) as a suitable method to have, at least, a reasonably good estimate of the relative stabilization gained in these compounds when the IMHB is formed. The isodesmic process we have used corresponds to the first reaction shown in Scheme 1.

Reaction (2) in Scheme 1, in which the carbon chain is fully deployed, was used just to check whether, in the isodesmic reactions proposed, the repulsive interaction between the terminal methyl groups may significantly affect the isodesmic energy obtained with the first reaction. The ideal situation should correspond to that in which the second reaction is thermoneutral. The results obtained for these two reactions at the G4 level of theory are summarized in Table 1. It can be observed that, indeed, the reactions (2) for the set of derivatives investigated are nearly thermoneutral, which is an indication of the reliability of our isodesmic reaction (1) to provide a good estimation of the stabilization energy associated with each IMHB. Not surprisingly, the largest deviation for thermoneutrality of reaction (2) is obtained for the first member of the series, as a consequence of its higher rigidity. Nevertheless, it should be taken into account that, in these isodesmic reactions, and, mainly, when the carbon chain is sufficiently large and very flexible, the number of possible conformers for the three compounds that do not have IMHB is very large, and, in many cases, the energy difference between them is very small, even smaller than 1 kJ·mol^−1^.

The values in Table 1 show, in agreement with the IMHB length reported in Figure 1, that the strongest isodesmic stabilization interaction energy corresponds to *n* = 2, being the weakest one corresponding to *n* = 0. These energetic trends are also consistent with the characteristics of the corresponding molecular graphs drawn in Figure 2a, showing that, in all cases, the electron density at the BCP located between the H atom of the hydroxyl group and the N atom of the amino group goes through a maximum for *n* = 2.

The 3D-NCI plots included in Figure 2b also show the existence of an isosurface between the OH and the NH_2_ groups, which denotes the existence of a NCI whose strength increases from *n* = 0 (greenish) up to *n* = 2 (blueish). It is also worth noting that, already for *n* = 1, a second greenish lobe appears attached to the blueish one associated to the van der Waals interactions range involving the chain of carbon atoms. Indeed, for *n* = 2, the two isosurfaces are now independent and, for the remaining systems (*n* = 3–5), the extension of these secondary interactions increases with the length of the chain of carbons.

The evolution of the characteristics of the IMHB along the series is nicely reflected in the IR spectra of the different species. As shown in Figure 3, the absorption band associated with the O–H stretching of the alcoholic function, which, for the first compound (*n* = 0), is predicted to appear at 3715 cm^−1^, is clearly redshifted when moving to larger compounds. This red-shifting is maximum (233 cm^−1^) for *n* = 2, but even for *n* = 5 the red-shifting with respect to *n* = 0 is significant (141 cm^−1^), indicating, in agreement with the other indices that the IMHB for *n* = 5, though weaker than that for *n* = 2 is still stronger than for *n* = 0.

### 3.2. HOCHX(CH_2_)_n_CH_2_NH_2_ and HOCH_2_(CH_2_)_n_CHXNH_2_ (n = 0–5, X = F, Cl, Br) Derivatives

Let us look now at the effects of halogen substituents at the α-position of both the -CH_2_OH and the -CH_2_NH_2_ functional groups. As indicated above, the number of possible conformers is huge, so, in Appendix A, we present only the optimized geometries of the most stable conformers stabilized by an O–H···N or a N–H···O IMHB. As it was also the case for the unsubstituted amino alcohols, and regardless of whether the halogen substituent is at the α-position of the CH_2_OH group or the CH_2_NH_2_ group, the conformers with a N–H···O IMHB are less stable than those with a O–H···N IMHB, with the only exception being the HOCH_2_CHXNH_2_ (X = F, Cl, Br) derivatives (see the first row of Appendix A), where the conformer with a N–H···O IMHB is predicted to be, at the G4-level of theory, slightly more stable than the conformer exhibiting a O–H···N IMHB regardless of the nature of the substituent X.

Focusing our attention then on the compounds stabilized by O–H···N IMHB, it can be observed that, as it was the case for the unsubstituted amino-alcohols, when the halogen substituent (F, Cl or Br) is at the α-position of the CH_2_OH group, the IMHB length decreases from *n* = 0 to *n* = 2, where it reaches its minimum value. An NBO analysis of these compounds not only indicates (see Figure 4a) that, as expected, the most significant orbital interactions involve the occupied lone pair of the amino group and the empty antibonding σ*_O–H_ orbital, which necessarily results in a weakening of the O–H bond, but also (see Figure 4b) that this interaction reaches its maximum, regardless of the nature of the substituent (F, Cl or Br) for *n* = 2. It is also very important to note that the same picture shows that this effect increases following the sequence: H < F < Cl < Br, parallel to the energy of the antibonding σ*_O–H_ orbital decreases, favoring the charge donation from the nitrogen lone-pair. Consistently, the electron densities at the O–H BCP decrease, but those at the IMHB BCP (See Figure 4c) increase following the sequence H < F < Cl < Br (see Figure 4c).

The effects on the IMHB characteristics by introducing the halogen substituent (F, Cl or Br) at the α-position of the CH_2_NH_2_ group are just the opposite as those just discussed for α-substitution with respect to the CH_2_OH group. Indeed, the presence of an electronegative atom at α-position of the CH_2_NH_2_ group results in a reduction of the intrinsic basicity of the amino group, which accordingly becomes a poorer donor towards the σ*_O–H_ orbital, leading to O–H···N IMHB weaker than in the unsubstituted compounds. Accordingly, this IMHB becomes about 10% longer. Again, the substituent effect increases as H < F < Cl < Br, and, therefore, whereas for substituents α to the CH_2_OH group the strongest O–H···N IMHB is observed for the Br derivative, when the substituent is α to the CH_2_NH_2_ group, the Br derivative exhibits the weaker IMHB. This behavior is in perfect agreement with the values of the electron densities at the BCP associated with the IMHB, as shown in Table 2 for the particular case *n* = 2 being taken as a suitable example.

These different trends are also nicely reflected in the characteristics of the corresponding IR spectra. We are going to illustrate this point again using the case *n* = 2 as a suitable example. As shown in Figure 5a, for the case in which the substitution takes place at the α-position with respect to the CH_2_OH group, the absorption band associated with the O–H stretching frequency is clearly red shifted upon F, Cl, and Br substitution. If the substitution takes place at the α-position with respect to the CH_2_NH_2_ group (see Figure 5b), the O–H stretching band is now blue shifted when H is replaced by F, Cl, Br.

### 3.3. Complexes between HOCH_2_(CH_2_)_n_CH_2_NH_2_ (n = 0–3) and BeF_2_

In this section, we will examine the effect that the interaction of the amino-alcohols HOCH_2_(CH_2_)_n_CH_2_NH_2_ (*n* = 0–3) with a strong Lewis acid as BeF_2_ will have on the IMHBs stabilizing these compounds and characterized in Section 3.1. However, the presence of the beryllium derivative opens up new scenarios in which new IMHBs enter into play as well as the possibility of having beryllium bonds replacing the IMHBs which characterize the isolated amino-alcohols.

Indeed, the association of BeF_2_ to the alcohol functional group (see the first column of Figure 6) leads to the amino-alcohols moiety structures being very similar to those exhibited by the isolated amino-alcohol, though the bond length of the OH···N IMHB is much shorter than the one found in Section 3.1 for the isolated system.

This reinforcement of the OH···N IMHB is the obvious consequence of the significant electron density transfer from the oxygen atom of the hydroxy group to the empty orbitals of the Be atom to form the corresponding beryllium bond, which necessarily increases the proton donor character of the OH group. Indeed, the second-order orbital interaction energies between the nitrogen lone-pair and the σ *_O–H_ antibonding orbital, responsible for the formation of the O–H···N IMHB, are about four times larger for the BeF_2_ complexes than for the isolated amino-alcohol as shown in Appendix A. The effect is qualitatively similar when BeF_2_ interacts with the amino group (second column of Figure 6), which, upon beryllium association, also becomes a stronger proton donor, reinforcing the N–H···O IMHB, but necessarily to a lesser extent than when the group involved in the alcohol function, with the only exception of the amino-ethanol, in which case the conformer with a N–H···O IMHB is predicted to be 9 kJ·mol^−1^ more stable than the one stabilized by the formation of a O–H···N IMHB (see Table 3).

It is also worth noting that, when the complex is stabilized through the formation of a N–H···O IMHB, the possible formation of a second IMHB by the interaction of the O–H group with one of the fluorine atoms attached to beryllium is open, and, indeed, as shown in Table 3, this new conformer is systematically (from 6 to 11 kJ·mol^−1^) more stable (compare the third and the fourth column of Table 3). Nevertheless, the most important finding is that, in all cases, the global minimum (fifth column of Table 3) corresponds to a complex in which no IMHBs are formed because the interaction of the O and N atoms of the amino-alcohol with beryllium atom of BeF_2_ molecule forming the corresponding beryllium bonds is energetically more favorable. The propensity of Be to exhibit a tetrahedral coordination has been previously reported in the literature when competing with intermolecular hydrogen bonds [47,70]. Here, we find that this tendency of Be to behave like a “tetrahedral proton” [47,70] is also observed when competing with IMHBs. In addition, importantly, the enthalpy gap between these bridged structures and the most stable conformers exhibiting a O–H···N IMHB is big enough (8.7 to 22.2 kJ·mol^−1^) as to conclude, using a Boltzmann distribution function, that, at room temperature, practically 100% of the complexes are those stabilized through the formation of beryllium bonds.

An inspection of the molecular graphs of these complexes (see Figure 7) confirms the tendency of Be to be tetracoordinated, the electron density at the N–Be beryllium bond being systematically larger than that at the O–Be beryllium bond, since the amino group is a better electron donor than the hydroxyl one. On the other hand, the intrinsic stability of the complex, measured by the corresponding binding energy, increases from the amino-ethanol to the amino-pentanol, but, for the amino-butanol, goes through a little sinkhole that is consistent with the fact that, as we have discussed before, the stabilization produced in the isolated compound by the O–H···N IMHB is maximum for amino-butanol.

Finally, it is interesting to highlight some of the peculiarities of the IR spectra of these complexes, showed in Figure 8.

The first conspicuous fact is that the stretching O–H absorption band appears significantly blue-shifted with respect to those in Figure 5a, as it corresponds to a free OH group that, in these systems, does not participate in hydrogen bonding. Indeed, the OH stretching frequency calculated for the bridge complexes in Figure 8 go from 3767 to 3832 cm^−1^, values rather similar to that obtained for methanol (3831 cm^−1^) at the same level of theory. However, for the amino-ethanol, the blue-shifting, with respect to the isomer in Figure 5a, is only 200 cm^−1^ because of the rather weak O–H···N IMHB in the isomer of Figure 5a. For amino-butanol and amino-pentanol, the shifting is greater than 1100 cm^−1^. Another common finger-print of the IR spectra of these complexes is the presence of a rather intense band always around 800 cm^−1^ corresponding to the symmetric stretching of the BeF_2_ moiety.

## 4. Conclusions

From our G4 calculations, we can conclude that, when F, Cl, or Br atoms are introduced as substituents at the α-position of both the hydroxyl or the amino group of amino-alcohols, there is a significant change in the strength of both the O–H···N or the N–H···O IMHBs that can be formed in these compounds. Substitution at the α-position of the hydroxyl group results in a significant reinforcement of the O–H···N, which follows the trend H < F < Cl < Br, the effects on the N–H···O IMHB being the opposite. Consistently, substitution at the α-position of the amino group also results in a reinforcement of the N–H···O IMHB, but the effect is weaker than the one affecting the O–H···N IMHB. Accordingly, the global minima for the unsubstituted and the substituted amino-alcohols are stabilized by the formation of O–H···N IMHBs. In addition, for both sets, the strength of the O–H···N IMHB, as shown by the characteristics of the corresponding molecular graphs, the NBO and NCI analyses, and the stabilization energy estimated through the use of isodesmic reactions, goes through a maximum for *n* = 2 (amino-butanol).

The scenario changes completely when the HOCH_2_(CH_2_)_n_CH_2_NH_2_ (*n* = 0–3) amino-alcohols interact with a typical electro-deficient compound such as BeF_2_. The interaction of BeF_2_ with either the hydroxyl or the amino group of the amino-alcohol changes the strength of the IMHBs dramatically. In the first case, the O–H···N IMHB becomes much stronger, whereas, in the second case, it is the N–H···O IMHB that becomes very much reinforced, though the effect is still weaker than the one observed for the O–H···N IMHB. In this second case, interaction of BeF_2_ with the amino group, a new possibility is open through the formation of an additional IMHB between the O–H group of the alcoholic function and one of the F atoms of the BeF_2_ molecule. However, it is the possibility for the beryllium atom to interact simultaneously with the O and the N atoms of the amino-alcohol that leads to the global minimum of the potential energy surface, with the result that the IMHBs are replaced by two beryllium bonds. These complexes stabilized by the formation of two beryllium bonds, O···Be and N···Be, present very peculiar IR spectra, in which the OH stretching band appears in some cases more than 1100 cm^−1^ blue-shifted with respect to the α-substituted amino-alcohols, whereas a very intense band, around 800 cm^−1^, corresponding to the symmetric stretching of the BeF_2_ moiety, is another finger-print of these spectra.

## Data Availability

The data presented in this study are available in the article and in the Appendix A.

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
