# Peer review of "Perturbating Intramolecular Hydrogen Bonds through Substituent Effects or Non-Covalent Interactions"

_molecules, 2021, doi:10.3390/molecules26123556_

Round 1
Reviewer 1 Report
In my opinion, the manuscript entitled "Perturbating Intramolecular hydrogen bonds through substituent effects or non-covalent interactions" (MOLECULES-1256860) by Al Mokhtar Lamsabhi, Otilia Mó, and Manuel Yáñez is a good example of a well-written report on an expertly carried out study – actually, it is hardly surprising as it has been carried out by the experts in the beryllium chemistry. In this particular case, authors used a high-level composite quantum-chemical method and different electron-density analysis tools including NBO-SOPT and AIM to investigate the interrelationships between the strength of the OH···N or NH···O intramolecular hydrogen bonds (IMHBs) and the type of halogen substituent in a series of amino-alcohols, HOCH2(CH2)nCH2NH2 (n = 0-5). According to the authors findings, the strength of the OH···N IMHB goes through a maximum for n = 2 and follows the trend H < F < Cl < Br , which is in full agreement with the corresponding infrared (IR) spectra. Another interesting result is that for amino-alcohols (n < 4) the interaction with BeF2 increases dramatically the strength of the IMHBs, and this is because the possibility for the beryllium atom to interact simultaneously with oxygen and nitrogen atoms leads to the global minimum of the potential energy surface, with the result that the IMHBs are replaced by two beryllium bonds. In general, all the calculations presented in this work seem to have been performed competently (especially the selection of the theory level, the second-order orbital interaction energy calculations within the framework of the NBO-SAPT donor-acceptor approach, etc. ) and the conclusions drawn are well supported by the obtained data and the experiment; actually, it is quite difficult for me to find any major weaknesses of the presented discussion and reasoning. Because of the excellent quality of the manuscript (concise abstract, extensive work undertaken together with the relevant results obtained, very good English – no typos found, etc.) my recommendation is to publish in MDPI Molecules as it is. Congratulations!
Author Response
Reviewer # 1
We are very grateful to our first reviewer for his/her encourage comments.
The reviewer did not ask for any change.
Reviewer 2 Report
“Perturbating Intramolecular hydrogen bonds through substituent effects or non-covalent interactions “ by Al Mokhtar Lamsabhi , Otilia Mó , Manuel Yáñez was done and written carefully. There is a large amount of authors' work, especially in the computational layer.
The high level ab initio method G4 was used to calculate the stability of selected conformers of 6 different unsubstituted amino alcohols [HO-CH2-(CH2)n-CH2-NH2] and three ones substituted with F, Cl or Br [OH-CHX-(CH2)n-NH2]. The most durable of the conformers of molecules showing an intramolecular hydrogen bond between the -OH and -NH2 end groups was taken into account, after a previous search of the conformational space with the use of the conformer-rotamer ensamble sampling of the CREST procedure. The hydrogen bond energy was calculated using the isodesmic method and compared with the values ​​of four differents approaches (QTAIM of electron density at the BCP located between the H atom of the hydroxyl group and the N atom of the amino group), electron localization function ELF, natural bond orbital analysis (NBO) and the noncovalent interaction formalism NCI. The hydrogen bond strength was compared with the calculated IR spectra of the analyzed compounds, confirming the conclusion about the strongest intramolecular hydrogen bond in 4-aminobutanol. The observed changes in the electron density due to the substitution of alpha and beta with halogen atoms were also confirmed by changes in the calculated infrared spectra.
A separate part of the study consists of calculations made for selected (aminoethanol, aminopropanol and aminobutanol) aminoalcohols substituted in the different positions the to OH group with beryllium difluoride. It has been shown that this molecule can form hydrogen bonds both through the beryllium atom itself interacting with the oxygen atom and the nitrogen atom, influencing their proton donor properties, but also through the interaction of fluorine atoms. t has also been found that the beryllium atom interact simultaneously with the O and the N atoms by permanently fusing the complex acting as a "tetrahedral proton". A competitive influence of this type of bonding with respect to those hydrogen bonds formed by the hydrogen atoms of the OH and NH groups was observed, also by calculating their infrared spectra.
Note 1. At this point, however (page 12, vv. 361-363), I would like to point out that the "shift" of the OH group stretching band to a higher frequency (approx.3750 cm-1) it is simply a band that comes from the free OH group in the molecules where the hydrogen atom does not participate in hydrogen bonding. This band should not be referred to as a "blue shift" (reserved for other phenomena) because it is simply a band derived from the free OH group in molecules where the hydrogen atom does not participate in hydrogen bonding.
Note 2 In the Introduction (pages 1 and 2) there is a very extensive discussion of the nature, strength, and methods of calculating the energy of intramolecular hydrogen bonding in general, but also in amino alcohols.
However, the noncovalent interactions with beryllium were mentioned only once, in addition, the reader was referred to the previous articles of the authors of the reviewed work (items 46, 47 and 50 of References).
The introduction should be extended by two or three sentences introducing the subject of unusual bonds of the beryllium atom, even at the cost of guesswork about typical HB bonds, widely known to organic chemists.
I believe that with the above corrections (Notes 1 and 2), the reviewed article “Perturbating Intramolecular hydrogen bonds through substituent effects or non-covalent interactions “ fully deserves publication in Molecules.
Author Response
Reviewer # 2
We are also very grateful to our second reviewer by his/her positive analysis of our work. The referee asks for two minor changes that we have addressed:
Note 1. At this point, however (page 12, vv. 361-363), I would like to point out that the "shift" of the OH group stretching band to a higher frequency (approx.3750 cm-1) it is simply a band that comes from the free OH group in the molecules where the hydrogen atom does not participate in hydrogen bonding. This band should not be referred to as a "blue shift" (reserved for other phenomena) because it is simply a band derived from the free OH group in molecules where the hydrogen atom does not participate in hydrogen bonding.
Our answer: We understand the viewpoint of the reviewer. It is true that in general this band cannot be designed as blue-shifted; but please note that what is said in this paragraph is “blue-shifted with respect to those in Figure 5a” which is absolutely correct. However, we agree with the reviewer that this band would not be blue-shifted with respect to a normal OH group, so we decided to slightly modify this paragraph to avoid any possible misunderstanding.
Note 2 In the Introduction (pages 1 and 2) there is a very extensive discussion of the nature, strength, and methods of calculating the energy of intramolecular hydrogen bonding in general, but also in amino alcohols.
However, the noncovalent interactions with beryllium were mentioned only once, in addition, the reader was referred to the previous articles of the authors of the reviewed work (items 46, 47 and 50 of References).
The introduction should be extended by two or three sentences introducing the subject of unusual bonds of the beryllium atom, even at the cost of guesswork about typical HB bonds, widely known to organic chemists.
Our answer: Following the suggestion of the reviewer we have added a small paragraph to provide further and useful information on beryllium bonds. As a consequence, five new references were included.
Reviewer 3 Report
Report on MS id: molecules-1256860
In this article Authors present comprehensive studies on the effect of substituent influencing the strength of intramolecular hydrogen bonds. The Introduction section gives potential reader a good explanation of the undertaken scientific problem. The level of calculations and the large spectrum of methods used by Authors for their purposes gives very solid base for high quality basic data production. Obtained in such a way numerical results from computational methods fully support discussion and final conclusions. The paper is well written. It is a rare situation when I have no any critical remarks or suggestion for Authors for the improvement of their submission. Therefore, I suggest publication of the paper by Al Mokhtar Lamsabhi, Otilia Mó and Manuel Yáñez as it stands.
Author Response
Reviewer # 3
We are also very grateful to our third reviewer for his/her positive comments.
The reviewer did not ask for any change.